# Avoidance of *Riba*-Based Loans and Enhancement of Quality of Life: An Indonesian Context Analysis

Romi Adetio Setiawan

Faculty of Islamic Economics and Business, Fatmawati Sukarno State Islamic University Bengkulu, Bengkulu 38211, Indonesia; romiadetio@mail.uinfasbengkulu.ac.id

**Abstract:** *Riba* (usury) has long been a contentious issue in Islam due to its adverse effects on economic equity and social wellbeing. This paper delves into the intricate relationship between refraining from the use of *riba*-based loans and the quality of life, with a specific focus on the unique context of Bengkulu, Sumatra, Indonesia. By conducting an extensive analysis of the existing literature and empirical evidence, this study explores the multifaceted dimensions of transitioning away from *riba*-based loans and their negative consequences. The findings demonstrate that the avoidance of *riba*-based loans leads to a ripple effect of positive changes and improved mental and physical wellbeing. Furthermore, the ethical dimension underscores the alignment of financial practices with an equitable society and moral values, thereby fostering religious awareness and realization. The paper argues that transitioning away from *riba*-based loans does not merely serve as a potent catalyst for improving the quality of life exclusively within Muslim communities but extends the impact, transforming the contemporary way of life into a more sustainable and inclusive financial ecosystem. This transformation is achieved by steadfastly prioritizing ethical conduct, spiritual fulfilment, social responsibility, and the equitable sharing of prosperity. This research provides valuable insights for policymakers, practitioners, and researchers who are dedicated to advancing the understanding and implementation of Islamic finance for the betterment of society.

**Keywords:** *Riba*-based loans; *Shari'ah*; social wellbeing; Indonesia; quality of life; ethics

## 1. Introduction

There is a consensus among Muslims that *riba*-based loans are deemed *haram* (impermissible) and abhorrent due to being unethical, attracting the disapproval and condemnation of Allah. As a result, Muslims are obligated to refrain from involving themselves in any usurious transactions (Ahmed 2014). Instead, loans from Islamic banks have garnered attention as a viable alternative to the conventional financial system, with its focus on promoting fairness, justice, and avoiding interest-based transactions (Azmat et al. 2020). The inherent values and principles of a non-*riba* loan extend beyond mere economic considerations, encompassing various dimensions that impact the quality of life for individuals and communities (Hassan 2017). These ethical loans not only rely on capital earnings and profits but also the adherence to *shari'ah* concepts. This approach seeks to achieve "*shari'ah* legitimacy" in their operations and business endeavours, aiming to fulfil the higher objectives of *shari'ah* and promote equitable social wellbeing (Ayub et al. 2023).

Despite the scepticism surrounding a non-*riba* loan, often dismissed as a replication of the conventional *riba* system, it is worth noting that an ethical financial system has invigorated the Muslim community and driven a religiously motivated transformation in its modern Islamic financial practices (Tok and Yesuf 2022; Farooq 2022; Khursheed et al. 2021). Islamic finance has been critiqued for its perceived commercialization of Islam, as noted by Roose (2020, p. 9):

> I consider how we are seeing the problematic compartmentalization of Islam, with Imams brought in as consultants and used to project an aura of Islamic

credibility, while having very little to do with the day to day operations of the business. This is in the coal face of the "silent revolution" and a space in which Islam becomes a product, a marketable commodity, to be bought and sold.

As argued previously, modern Islamic financial institutions play a significant role in cultivating piety within the Muslim community. This viewpoint is consistent with Kaakeh's observation who noted, "As Islam prohibits all kinds of interest, Muslims seek to obey their religion and therefore deal with Islamic banks as an acceptable alternative to conventional banks"(Kaakeh et al. 2018, p. 218). Proponents of Islamic finance endeavour to revitalize contemporary society during times of crisis through these institutions. This process is aptly described by Ali (2023, p. 1) as "Re-introducing Islam to Muslims-proselytization".

The reason why this research focuses on the distinctive context of Bengkulu, Indonesia, is that, based on Google Trends (2023), public interest in Islamic economics in Bengkulu has been growing steadily in the past twelve months; it ranks number three out of the top five subregions in Indonesia. Bengkulu is a province located on the southwestern coast of Sumatra, Indonesia. It is known for its rich cultural heritage, lush landscapes, and historical significance. The province is a blend of urban and rural areas, with its capital city, also named Bengkulu, serving as the economic and administrative centre. Bengkulu's economy is diverse, encompassing agriculture, fishing, mining, and trade. In recent years, there has been a growing interest in Islamic finance, as evidenced by the presence and growth of Islamic banks and Shariah Rural Banks (SRBs) in the region (BPS-Statistics of Bengkulu Province 2023). Bengkulu provides an interesting case study due to its predominantly Muslim population with 97.67% as of March 2023 (Population and Civil Registration Agency 2023) and the presence of a thriving *shari'ah* modes of financing including Islamic banks, *Baitul Maal Wa Tamwil* (BMT), and *shari'ah* cooperatives.

As of 2023, notable financial institutions operating in Indonesia encompass Bank Mandiri, Bank Rakyat Indonesia (BRI), Bank Central Asia (BCA), and Bank Negara Indonesia (BNI), each exercising a significant presence and robust network within the region of Bengkulu (Bank Indonesia 2022). In the realm of Islamic finance, Bengkulu accommodates a portfolio of five Islamic banks, namely Bank Syariah Indonesia, Bank Muamalat Indonesia, Bank Mega Syariah, Bank Sinarmas Syariah, and Bank Tabungan Negara Syariah. Additionally, there exist three Shariah Rural Banks (SRBs) in this locale, identified as SRB Maslahat, SRB Fadhilah, and SRB Muamalat Harkat (Bank Indonesia 2022). According to Bank Indonesia (2022), as per November 2022, the total assets of Islamic Banks in Bengkulu amounted to IDR 2373 trillion. This figure, while substantial, remains considerably lower than the assets held by conventional banks, which stood at IDR 30.3 trillion during the same period. However, it is noteworthy that there is a growing interest among the populace in Islamic banking as compared to conventional banking. The assets of Islamic banks have exhibited an impressive year-on-year growth rate of 18.29%, surpassing the 8.12% year-on-year growth rate observed in conventional banks. This trend underscores a rising preference for Islamic financial services within the region (Bank Indonesia 2022).

Here, the Muslim community of Bengkulu has inherently embraced and practiced s*hari'ah*-compliant commercial transaction in their traditions, often without explicit awareness that their actions align closely with the principles of *shari'ah*. This paper sheds light on the profound influence of the s*hari'ah* mode of financing in Bengkulu, facilitating a comprehensive and robust process of Islamization within traditional practices. Consequently, this process leads to heightened religious awareness and realization among the populace.

By examining the multidimensional aspect of the *shari'ah* mode of financing and its contribution to societal development, this study aims to illuminate the transformative potential of the non-*riba* loan system. To achieve this objective, the study employs both sociological and anthropological approaches to comprehensively examine the multifaceted dimensions of the *shari'ah* mode of financing in Bengkulu, investigating how it enhances the quality of life within Muslim communities. Central themes such as ethical conduct, spiritual fulfilment, social responsibility, and shared prosperity are examined to discern the profound impact of Islamic ethical business on society. By illuminating the unique context

of Bengkulu and its experiences with the non-*riba* loan system, this research contributes to a deeper understanding of how modern Islamic financial system can serve as catalysts for positive change, fostering a more equitable and inclusive society rooted in Islamic values.

## 2. Literature Review and Study Context

A variety of academic studies have explored different aspects of *riba* (interest) in the financial system, such as its impact on economic growth (Iswanaji 2018; Iqbal and Mulyaningsih 2019; Bitar et al. 2021), financial stability (Siswantoro 2022; Effendi 2013), and customer satisfaction (Muallim 2003; Hoq et al. 2010; Khursheed et al. 2021). However, there has been relatively less research conducted on the relationship between avoiding *riba* and economic wellbeing or quality of life (Waysith 2017; Iqbal and Mulyaningsih 2019). For instance, Waysith (2017) investigated how Islamic microfinance positively influenced the quality of life for low-income individuals by reducing poverty. Similarly, Iqbal and Mulyaningsih (2019) highlighted the positive impact of non-*riba* loans on improving the quality of life for poor and vulnerable households through increased access to financial services.

Furthermore, (Soemitra et al. 2022); (Khursheed et al. 2021) explored the contribution of *shari'ah* finance institutions to social welfare and community development, particularly through mechanisms like *waqf* (endowment) and Islamic social finance.

In contrast, this study offers a unique perspective on the relationship between avoiding *riba*-based loans and quality of life, from multidimensional aspects focusing specifically on Bengkulu city in Sumatra, Indonesia. It examines how *shari'ah*-mode financing institutions in Bengkulu foster a comprehensive and robust process of shaping individuals' understanding of *riba* in Islamic transactions and influence their engagement with religious, social, and cultural practices related to economic wellbeing. To gain insights into the impact of non-*riba* loans on the quality of life in this localized context, the study considers various aspects of the community's daily lives, including their behaviour, interactions, and personal and social discourses regarding non-*riba* transactions. The aim is to contribute to the existing knowledge on *shari'ah*-compliant finance and its societal impact. The findings of this research hold implications for policymakers, financial institutions, and researchers aiming to promote sustainable development, social inclusivity, and ethical financial practices not only in Bengkulu but also in other relevant contexts.

## 3. Methodology

This research utilizes qualitative research methods to explore the intricate facets of transitioning away from interest-based loans, known as *riba*, and its contributions to societal advancement within the context of Bengkulu. Qualitative research is a methodological approach employed to acquire profound insights into human behavior, attitudes, beliefs, and experiences (Alamri 2019). In the context of this study, the use of qualitative methods is indispensable, as it allows researchers to explore the topic by engaging directly with participants where they can express their thoughts and feelings openly. Data for this study were meticulously gathered through an extensive multi-site fieldwork initiative conducted between January 2020 and January 2023. In-depth interviews were conducted with individuals who had accessed credit from non-*riba* Islamic financial institutions, ranging from 45 min to 1.5 h each. Some interviews were conducted in a mosque, after the prayer providing a conducive atmosphere to open conversation. Other interviews occurred in private meeting rooms at community centres, allowing for a confidential and secure environment for participants to share their experiences.

To capture the richness of the discussions and ensure accuracy in data analysis, all interviews were tape recorded with the participants' consent. High-quality digital audio recorders were used to capture the conversations in their entirety. These recordings were then transcribed shortly after each interview to maintain the fidelity of the participants' responses and to facilitate subsequent data analysis. The narratives extracted from these interviews were employed to dissect cases related to the deliberate avoidance of *riba* and its repercussions on the quality of life in Bengkulu, Indonesia.

The overarching objective of this investigation is to analyze and synthesize the available data to construct a comprehensive understanding of the subject matter. The data, drawn from both the extensive literature review and empirical studies, were subjected to qualitative analysis. This qualitative approach, in conjunction with a thorough examination of the existing literature and empirical evidence, contributes to a well-rounded comprehension of the multifaceted impact of refraining from *riba* on the quality of life in Bengkulu, Indonesia.

In the broader context of contemporary society, characterized by the widespread adoption of conventional banking practices grounded in capitalist principles, there exists a potential for the erosion of traditional and religious norms among Muslims. This study undertakes an exploration of the implications associated with Muslims opting for non-*riba* financial credit as a means of preserving their religious devotion within the traditional Muslim community situated in Bengkulu, Indonesia. Furthermore, this research closely examines the individual behaviour in question, placing it under scrutiny for its repercussions on interpersonal relationships, particularly within the context of father–son dynamics. It is posited that these interpersonal dynamics, in turn, exert a discernible influence on family structures and daily life within the Muslim community.

This article relies on testimonies provided by eleven participants who reflect on their experiences with non-*riba* financial credit. Additionally, insights from four participants who engage with both conventional financial institutions and Islamic financial institutions in their business operations are considered for comprehensive analysis. The participants in this research were randomly selected from different age groups as well as the community leaders from 10 districts of Bengkulu. See Table 1.

**Table 1.** Profile of interviewees in Bengkulu, Sumatra, Indonesia.

| No | Category of Participants | Age | Education | Occupation |
|----|--------------------------|-----|-----------|------------|
| I | Participants who reflected on their experience with Islamic finance | | | |
| 1. | Astri | 44 | Senior High School | Homemaker |
| 2. | Herman | 47 | Bachelor's degree | Teacher |
| 3 | Zahra | 52 | Elementary School | Housewife |
| 4 | Zainal | 45 | Bachelor's degree | Trader |
| 5 | Rida | 55 | Elementary School | Tailor |
| 6 | Syakrani | 68 | Diploma 3 | Community leader (*Pak RT*) |
| 7 | Nurul | 54 | Bachelor's degree | Businessman |
| 8 | Hadi | 38 | Senior High School | Fisherman |
| 9 | Aan | 28 | Bachelor's degree | Trader |
| 10 | Fenti | 27 | Bachelor's degree | Teacher |
| 11 | Fadli | 50 | Elementary School | Trader |
| Total number of participants | | | | 11 |
| II | Participants who engaged in both Islamic and Conventional Banks | | | |
| 1. | Andi | 33 | Master's degree | Government civil servant |
| 2. | Ahmad | 39 | Bachelor's degree | Government civil servant |
| 3. | Dedi | 40 | Master's degree | Government civil servant |
| 4 | Faisal | 37 | Bachelor's degree | Teacher |
| Total | | | | 4 |

Source: data produced by the author.

## 4. Dimensions of *Shari'ah* Mode Financing and Quality of Life

In a recent investigation conducted by Ayub et al. (2023), it was established that *riba* (interest) yields detrimental effects on society, whereas ethical finance offers unique dimensions to enhance economic wellbeing. Notably, research has shown that Islamic

banking establishments have played a role in promoting financial stability and resilience, particularly during times of global financial upheavals (Alrifai 2015; Bhatti and Bhatti 2009; Reno and Abdullah 2016; Setiawan 2018). The principles of risk-sharing, avoidance of interest (*riba*), and asset-based financing in non-*riba* loan systems have contributed to increased financial inclusivity, providing access to financing for small and medium-sized enterprises (SMEs) and individuals who were previously underserved (Shabsigh et al. 2017). As the problems of poverty and socioeconomic injustice are multidimensional, Islamic microfinance, through institutions such as Islamic microfinance banks and Islamic cooperative societies, has also played a significant role in poverty alleviation and improving livelihoods in various countries under the system of *awqaf* (donated asset) for socioeconomic and broad-based welfare (Effendi 2013; Ayub et al. 2023). Additionally, the *Shari'ah*-mode financing system promotes responsible investment and discourages making a profit from speculation (*mujazafat*), which contributes to sustainable economic growth (Setiawan 2023). Beyond its economic implications, *shari'ah* financing has significant potential to influence the overall wellbeing and quality of life of individuals in various dimensions.

One of the key dimensions of non-*riba* loan system lies on its emphasis on social justice and ethical conduct. *Shari'ah*-mode financing institutions have developed products and services, such as Islamic microfinance, which cater for the needs of marginalized communities and provides accessible loan options for individuals with limited access to traditional banking services (Abul and Sabur 2018; Effendi 2013). Furthermore, these institutions encourage the collection and distribution of *zakat* (obligatory alms) and the establishment of *waqf* (endowment) funds, which are directed towards addressing social welfare needs, such as poverty alleviation, education, and healthcare (Triyowati et al. 2018). Thus, s*hari'ah*-compliant financing is not designed for mere profit making but also to encourage social wellbeing.

Adhering to a *riba*-free framework within the financial system also serves to bolster the emphasis on financial inclusion and empowerment. This system is based on equitable financing and profit sharing, making it particularly well-suited for aspiring small-scale entrepreneurs (Effendi 2013). Through the provision of interest-free loans and entrepreneurial support, coupled with an endorsement of profit sharing (*mudharabah)*, this approach underscores the ethical use of capital. As highlighted by Qomar (2018), it actively encourages investments in environmentally friendly projects and ventures. Chong and Liu (2009) have further observed that the *shari'ah*-compliant financial institutions often prioritize empowering individuals to start businesses, improve their economic status, and lead more dignified lives.

Furthermore, a non-*riba* financing system recognizes the spiritual dimension of individuals' lives and seeks to align financial activities with Islamic values. It promotes the concept of *halal* (permissible), ethical business, transparency, and accountability in financial transactions, fostering trust and integrity (Wilson 1997). Wijiharta et al. (2022) assert that by adhering to principles that prohibit interest-based transactions and promote ethical investment, the *Shari'ah*-mode financing system provides individuals with the opportunity to align their financial activities with their religious beliefs, which can contribute to spiritual fulfilment and a sense of inner peace and happiness.

In what follows, we see that the *Shari'ah*-mode financing institutions in Bengkulu, Indonesia, transcend mere financial transactions, demonstrating the profound dimensions of non-*riba* loans. The adherence to *Shari'ah*-mode financing enables societies in Bengkulu to uphold piety at the individual level, shaping the relationship between Islamic financial institutions and quality of life within the Muslim community akin to that of *a father and a son*.

## 5. The Multidimensional Node and Its Anticipated Roles and Outcomes

During my fieldwork in Bengkulu, I had a conversation with one of my informants, and asked her, "do you know about the free-interest loan system in Islamic banking?". Her response was rather surprising, "I don't know yet if there is an Islamic banking system,

never heard about it, I only know the system of a loan of money from a bank but not that kind of interest-free system" (interview with Astri, Bengkulu, 12 December 2022). Similar answers were also given by some of my other research subjects, stating that they have taken a loan from lenders (*rentenir)* who charge interest.

This existing schism pertaining to the dearth of s*hari'ah-*mode financing literacy serves as a tangible manifestation of the significant challenges faced by Islamic banking in Indonesia, particularly concerning the multiplicity of perceptions prevalent in various subregions across the country (Ilyana et al. 2022). The principle of reciprocity entails an individual making a well-informed decision to utilize a product or service only after thoroughly exploring and comprehending its features and attributes (Bhabha et al. 2014). The Islamization of economic and financial industries in Indonesia is part of the government's national project, evident from the establishment of Komite Nasional Ekonomi dan Keuangan Syariah (The National Islamic Finance Committee) (KNEKS) at the national level and Komite Daerah Ekonomi dan Keuangan Syariah (The Regional Islamic Finance Committee) (KDEKS) for regional areas. Herman shared his thoughts about *riba* literacy during my interview:

> We need to raise public awareness about the concept of interest-free in Islamic finance; this is the role of academics. They need to write easily understandable articles to educate the public through various channels such as social media, newspapers, and other platforms. (interview with Herman, Bengkulu, 4 June 2023)

In this sense, a non-*riba* financial system represents a transformative model that signifies a transition from traditional modes of financing to a contemporary approach guided by Islamic principles (Homoud 1994). Despite the earnest efforts of the government and Islamic bank to promote non-*riba* loan literacy, a considerable number of Indonesians remain uninformed about its principles. This knowledge gap has led to disjuncture and polarization within society. It is crucial to recognize that s*hari'ah-*mode financial institutions move beyond being profit making; they also serve as a platform for proselytization, focusing on educating individuals about the religious aspects of finance. In conversations with informants in Bengkulu, numerous informants expressed their expectation that the s*hari'ah-*mode financing system would align with their beliefs, inspiring them to exclusively participate in transactions deemed permissible (*halal*). As articulated by Zahra, a fifty-two-year-old woman from Kampung Bahari village, "After utilizing a non-*riba* loan, I feel a stronger sense of piety and have become more cautious about avoiding prohibited transactions" (interview with Zahra, Kampung Bahari, 10 January 2023). The image portrays a robust bond between Islamic financial institutions and the individuals' spiritual journey, fostering a sense of unity akin to a close-knit family. The idea of this connection enhances their social prestige and the inherent legitimacy that accompanies it (Farooq 2022).

Another interviewee such as Zainal also said that *shari'ah-*compliant financing provided by Islamic financial institutions significantly helps in settling the capital problem faced by his business. His fish trading activity experiences rapid turnover of capital on a daily basis, with customers often paying for their purchases at a later date. Consequently, Zainal faces a substantial number of outstanding payments. To address this issue, he utilizes *Shari'ah-*compliant financing as additional capital, enabling his business to thrive and expand (interview with Zainal, Bengkulu, 10 January 2023). During discussions concerning the attainment of piety through Islamic finance, my informant frequently emphasized the transformative impact of comprehending Islamic financial institutions. He described how this understanding altered his spiritual journey, leading to behavioural changes that reflected a heightened awareness and adherence to Islamic principles in his business activities (interview with Zainal, Bengkulu, 10 January 2023). Another informant such as Rida a fifty-five-year-old lady, a tailor in Bengkulu, shared her experience of religious awareness:

> Before understanding *Shari'ah*-compliant financing, I was ignorant about the accessibility of capital, as I considered both *Shari'ah*-compliant and conventional capital to be the same. However, after gaining knowledge, I became aware that in our transactions, we must consider the *Shari'ah* aspect and differentiate between

*Shari'ah*-compliant and conventional capital, while avoiding any involvement with interest (*riba*)... and should not cheat, be ethical, and should stay on the right path. (interview with Rida, Bengkulu, 10 January 2023)

Rida's view on the experience of avoiding *riba* after acknowledging the Islamic finance was shared by other informants in Bengkulu. They frequently expressed their aspirations to embrace an authentic Islamic way of life, and the observable evidence indicates their earnest efforts to conduct business in a manner consistent with their beliefs (cf. Wilson 1997). For example, Syakrani, one of the community leaders (Pak RT) in Bengkulu, asserted:

I previously had no knowledge of *Shari'ah*-compliant financial systems. We used to provide loans to individuals in urgent need from the communal cash fund collected during each regular social gathering (*arisan*). If we had known about it earlier, perhaps we wouldn't have used interest in our borrowing and lending system. (interview with Syakrani, Bengkulu, 7 January 2023)

Their awareness of Islamic finance as an ethical approach to conducting business motivated them to engage in various religious acts. These included extending hospitality and providing alms to the less fortunate through Islamic financial institutions. The accessibility of s*hari'ah*-mode financing fosters the cultivation of piety both at the individual level and within the broader institution of the Muslim community.

The awareness of avoiding *riba*-based loans can be seen as one of the many approaches aimed at reducing habits that are detrimental to society, while concurrently expanding its influence and fostering the creation of authentic Islamic finance products that not only yield profitability but also contribute to socioeconomic development. This process involves navigating a complex network of relationships, often characterized as a "multiplicity of nodes" (Van Der Veer 2002, p. 96). In this sense, the establishment of closer ties with Islamic financial institutions can be metaphorically represented as the "father" node, while the strengthened connections with community members in disparate locations are analogous to the "son" node.

For the Muslim community in Bengkulu, understanding *riba* brings them to a purely aesthetic perception of *shari'ah*, which incurs a demand for s*hari'ah*-compliant financing as an embodiment of the Islamic way of life. This is evidenced through the sentiments expressed by Syakrani, a local community leader (Pak RT):

In our village, we have always been collecting funeral contributions every three months for providing assistance to the bereaved family in the event of a death tragedy, such as purchasing burial shrouds, digging graves, and conducting funeral ceremonies. (interview with Syakrani, Bengkulu, 10 January 2023)

This type of funeral contribution aligns with the concept of *takaful* (mutual cooperation) in *shari'ah* principles, promoting mutual cooperation and assistance (Sang et al. 2020). In Islamic finance, this system has evolved further, enabling *takaful* funds to be actively managed and invested in *halal* (permissible) ventures, thereby generating profits (Maf'ula and Mi'raj 2022). The remaining participants expressed astonishment upon discovering that numerous financial practices within society align with *Shari'ah* principles (interview with Hadi, Aan, Fenti, Fadli, Bengkulu, 10 January 2023). In the context of modernity, where capitalism's imperatives prevail and potentially pose challenges to traditional and religious financial systems, the emergence of the *Shari'ah*-compliant financial system offers devout Muslims a viable alternative that aligns with their religious beliefs. As described by Schrader (2000, p. 39) "Islamic banking is not only an inner struggle between traditional and modernist values, but a reaction to the Western, disembedded market model."

The uncertainty surrounding whether their traditional activities align with the *Shari'ah*-compliant financial system is also evident in Faisal's statement:

There's still a lack of awareness about *riba* in our community. This underscores the importance of ongoing efforts to educate people about the essence of *Shari'ah*-

compliant financing. Let's persevere in these initiatives to promote knowledge and understanding. (interview with Faisal, Bengkulu, 10 January 2023)

One of the interviewees elaborated on their experience with forward selling activities related to Durian fruits. However, there is uncertainty regarding the conformity of this practice with Islamic principles, as demonstrated by the following statement: "Usually, we sell durians in bulk. Buyers purchase all the durians from our trees by estimating the total number of visible fruits on the trees. It's our customary practice. However, we're unsure about the Islamic perspective on this practice (interview with Fadli, Bengkulu, 10 January 2023)".

## 6. Balancing Commitment to and Deviation from *Shari'ah* Values

Although the comprehension of *riba* (interest) has expanded in specific regions of Indonesia, there remains a notable segment of the population, comprising individuals who identify as Muslims, who remain hesitant or resistant to fully embracing the concept (Riza 2018). This section delves into the concrete challenge faced by practitioners in striking a delicate balance between upholding the fundamental principles and guidelines of *shari'ah*-compliant financial institutions and, simultaneously, embracing essential adaptations and innovations to meet the diverse needs of their customer base.

In practice, the Islamization of financial institutions to cater for the needs of Muslims can potentially pose a challenge that leads to the development of Islamic-finance phobia within certain segments of society. This phobia can emerge even in some Muslims who may have misconceptions or misunderstandings about *riba* or hold negative perceptions about *shari'ah*. This phenomenon is particularly pronounced when fervent proponents contend that there is no distinction between Islamic banks and conventional banks, asserting that both entities engage in *riba* (interest) for their profits. One of the examples is the discourse related to the interest system of a conventional bank as *riba* by the Indonesian Ulema Council (Majelis Ulama Indonesia-MUI). Some hold negative perceptions, like Sarwat (2023, p. 1), for example, the owner of rumahfiqih.com, who wrote "Actually, the main issue is not the size of the interest or profit-sharing, but rather the usage of the term "profit-sharing" as a camouflage for interest. In essence, it is still interest, but labelled as profit-sharing. These are the major "sins" of Islamic banks".

Achieving the right balance between commitment and deviation is essential for the sustainable growth and development of *shari'ah*-mode financing. In my experience, in Bengkulu, there were several people who were sceptical and had a negative experience with Islamic financial institutions. For example, Nurul, a businessman, shared his negative experience:

> Actually, the problem is not just about the religious rulings against interest (*riba*), but there are other temptations in the form of profit-sharing systems that look like interest without risks for investors, and the profit amount is decided in advance, even before knowing the total profit. Sometimes these factors make me feel negative and create a strong doubt or scepticism (*su'udzon*) regarding Islamic banking. (interview with Nurul, Bengkulu, 20 January 2023)

For Nurul, committing to the exclusive refraining from *riba* in his daily life was difficult, primarily due to the complexities and diverse interactions he faced in his business activities. As a result, he found himself compelled to utilize both conventional banks and Islamic banks. Despite this situation, Nurul firmly believed in striving to be a good Muslim and adhering to the teachings of Islam in his daily life, avoiding any involvement in prohibited (*haram)* activities. While he acknowledged the practical difficulties of relying solely on Islamic financial institutions, he remained committed to integrating Islamic principles into his lifestyle to the best of his ability, demonstrated by this statement: "Even though I still use an account in a conventional bank for my business, in my daily life, I strive to remain faithful to Islamic teachings, especially when it comes to avoiding any prohibited actions" (Interview with Nurul, Bengkulu, 20 January 2023). Other informants, particularly

civil servants, also provided similar examples where they maintained two separate bank accounts. One account was held in a conventional bank for receiving their salaries, while the other account was with an Islamic bank for deposit purposes (interview with, Andi, Ahmad, Dedi, Bengkulu, 19 December 2022). As Nurul succinctly articulated, "In the present day, it is challenging for everyone to manage with just a single Islamic bank account, given the business demands that necessitate the use of multiple bank accounts (interview with Nurul, Bengkulu, 20 January 2020)".

It is important to recognize that abstaining from *riba* raises complexities in its implementation and is subject to ongoing debates and discussions. Achieving a balance between the commitment to foundational principles and necessary adaptations is an ever-changing and dynamic process. As the modern Islamic financial industry evolves and responds to shifting economic and societal conditions, opinions on this matter may continue to evolve as well. Nonetheless, maintaining this balance is of utmost importance to safeguard the authenticity and integrity of s*hari'ah*-compliant financing, while concurrently ensuring its sustained relevance and effectiveness amid the constantly evolving financial landscape.

## 7. Conclusions

This paper highlights the multifaceted dimensions of transitioning away from *riba*-based loans and its profound impact on the quality of life within the context of the Muslim community in the region of Bengkulu, Indonesia. The findings reveal that the avoidance of *riba*-based loans contribute to enhanced spiritual devotion, economic equity, and ethical conduct. Furthermore, the ethical dimension of non-*riba* loans aligns with Islamic principles and values, promoting social responsibility and community welfare. This alignment leads to heightened religious awareness and realization among individuals, facilitating a deeper connection with their faith and cultural heritage. Furthermore, the emergence of s*hari'ah*-compliant finance serves as a response to the challenges posed by modern capitalism, offering devout Muslims a means to navigate the modern financial landscape while upholding their religious principles. This stronger connections and partnerships with Islamic financial institutions can be likened to a "father and son" node in a metaphorical sense, which then impacts the family.

However, there are challenges in balancing the commitment to Islamic principles while accommodating the necessary adaptations to meet diverse customer needs. Some individuals may be hesitant to fully avoid *riba* due to misconceptions or negative perceptions, especially concerning the differentiation between Islamic and conventional banks. Striking the right balance between adherence to Islamic principles and meeting modern demands is crucial for the sustainable growth of Islamic finance. This article highlights the importance of addressing the lack of awareness of *riba* within society to overcome the Islamic-finance phobia that may exist in certain segments. The transformative potential of a non-*riba* loan is evident in its influence on individuals' spiritual journeys and its contribution to their overall wellbeing and quality of life.

In conclusion, the multifaceted dimensions of s*hari'ah* mode financing extend beyond the realm of mere profit-making transactions. It is imperative to strike a delicate between commitment and deviation in various aspects, including ethical conduct, social responsibility, financial inclusion, and spiritual fulfilment. This balance is crucial for the long-term success of their noble goal, known as "*Shari'ah* Legitimacy" of these industries. Regular socialization and education initiatives are necessary to enlighten individuals about the essence of s*hari'ah*-compliant finance and its alignment with Islamic principles, fostering greater understanding and acceptance within society. Overall, the journey toward a more equitable and just financial system is ongoing, and further research in this area will undoubtedly contribute to advancing this important endeavour.

**Funding:** This research received no external funding, and the APC for this special issue is waived by MDPI.

**Institutional Review Board Statement:** This study was approved by the Institutional Review Board of the Fatmawati State Islamic University of Bengkulu, Indonesia. Institutional Review Board Code: 800 /Un.23/L.J/TL.01/10/2020 Approval Date: 3 Years (January 2020–January 2022) and extended up to January 2023.

**Informed Consent Statement:** Informed consent was obtained from all subjects involved in this study.

**Data Availability Statement:** Data is unavailable due to ethical restrictions.

**Conflicts of Interest:** The author declares no conflict of interest.

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
