# Peer review of "Avoidance of Riba-Based Loans and Enhancement of Quality of Life: An Indonesian Context Analysis"

_religions, doi:10.3390/rel14111376_

Round 1

Reviewer 1 Report

Comments and Suggestions for Authors

The topic that the author addresses is important, and the research design is clear. However, the analysis does not meet the initial expectations. The text has several weaknesses. Firstly, the author fails to provide vital information regarding the socio-demographic composition, socio-economic characteristics, distribution, and size of financial institutions in Bengkulu, Sumatra, the specific case being studied. Secondly, there are doubts about the methodology used, given the inconsistency in the number of stakeholders interviewed and the absence of socio-biographical profiles of the interviewees. Finally, the author seems to present her/his conviction rather than argue through trial and error a research hypothesis, which is the ethical superiority and economic effectiveness of Islamic finance over interest-based loans. This superiority needs to be demonstrated, and the case study could have provided valid arguments for the thesis, arguments which are not developed satisfactorily.

Author Response

Dear Reviewer,

Kindly find the detailed analysis and point-by-point response in the attached file.

Thank you for your time and consideration.

Reviewer 2 Report

Comments and Suggestions for Authors

The author uses mainly few interviews conducted in Bengkulu, in Indonesia to asses the positive effects of non Riba-Based Loans. The topic is original since  :1. the effects of Islamic finance on the lives of people is under researched. 2. focus on Bengkulu. For this, the research addresses a gap in the field. The interviews add a unique perspective of the population of Bengkulu as they struggle in achieving balance between conventional finance and Islamic finance.

However, beside English which needs extensive editing (meaning it should be proofread by a native speaker or a professional), the methodology is lacking. First, a section is needed to describe the profiles of the interviewees and the context in Bengkulu. Then, the selected quotes from the interviews should be separated from the analysis. The method selected for analysis should be clarified in the introduction (what approach does the author represent within the various sociological and anthropological theories?). The author cannot be part of the producers of discourse like in page 5 quoting himself/herself. The author observes and reports the ongoing dynamics with a „cold eye”. Also, the author makes several normative comments as on page 6 „This observation highlights the prevailing lack of awareness regarding riba within the community. Consequently, it underscores the importance of conducting regular socialization initiatives to enlighten individuals about the essence of Shari’ah-compliant financing, which extends beyond mere profit-seeking to encompass principles of mutual assistance, profit sharing, and ethical conduct in business dealings.” This makes the study inappropriate for a scientific journal that describes and discusses given dynamics.

Comments on the Quality of English Language

Extensive editing of English language required. Otherwise, this article is unclear.

Author Response

(The authors gave the same response as above.)

Round 2

Reviewer 1 Report

Comments and Suggestions for Authors

The author revised the text satisfactorily according to the referee's observations and suggestions. The article is now richer in information, improved in the methodological part and in the main arguments of the analysis of the case studied.

Author Response

I highly appreciate your positive assessment of the changes made in my manuscript revision. I am delighted to hear that you found the revised text to be satisfactory and that it has contributed to enhancing the richness of information, improving the methodological aspects, and strengthening the main arguments in the analysis of the studied case.

Your feedback has a significant impact on refining the content, and it encourages me to continue striving for excellence in the research and writing.

Thank you for your support.

Reviewer 2 Report

Comments and Suggestions for Authors

Methodology has improved. However, I recommend the extensive editing of English language before accepting the article for publication.

Comments on the Quality of English Language

Methodology has improved. However, I recommend the extensive editing of English language before accepting the article for publication.

Author Response

I am pleased to hear that you found the improvements made in the methodology to be satisfactory.

I also appreciate your recommendation regarding the extensive editing of the English language in the article before publication. To address this issue, I have utilized the services provided by MDPI English Editing. This service involves a thorough editing process carried out by either a skilled native speaker or a professional editor. Furthermore, I have submitted the English Editing Certificate along with the manuscript.